# A Review of the Development of Key Technologies for Offshore Wind Power in China

**Qixiang Fan** [1,*], **Xin Wang** [2], **Jing Yuan** [2], **Xin Liu** [3], **Hao Hu** [4] **and Peng Lin** [2,*]

1. China Huaneng Group Co., Ltd., Beijing 100031, China
2. Department of Hydraulic Engineering, Tsinghua University, Beijing 100084, China; xin-wang19@mails.tsinghua.edu.cn (X.W.); yuanj2021@mail.tsinghua.edu.cn (J.Y.)
3. Huaneng Clean Energy Technology Research Institute Co., Ltd., Beijing 102209, China; x_liu@qny.chng.com.cn
4. Huaneng Jiangsu Clean Energy Branch, Nanjing 210008, China; huhao3232@126.com
* Correspondence: qx_fan@chng.com.cn (Q.F.); celinpe@tsinghua.edu.cn (P.L.)

**Abstract:** In recent years, Offshore Wind Power (OWP) has gained prominence in China's national energy strategy. However, the levelized cost of electricity (LCoE) of wind power must be further reduced to match the average wholesale price. The cost-cutting and revenue-generating potential of offshore wind generation depends on technological innovation. The most recent studies and applications of offshore wind technology are thoroughly examined. (1) Techniques for site selection, such as site surveys, wind resource assessments, and environmental factors, are reviewed. (2) Three main technical components in offshore wind farms are discussed, including wind turbine, foundations, and booster stations. (3) The state-of-the-art method of the offshore wind farm's construction and operation and maintenance (O&M) practices is discussed. In situ marine geological surveying, large-scale offshore wind turbine manufacturing, integrated structural design, floating foundation design, flexible DC transmission technology, shortage of specialized vessels and equipment for construction, intelligence of O&M, and other issues are challenging China's OWP industry. A brief overview of China's efforts in standardization, parity, and research and development are discussed. Recommendations for future development of the wind power industry are provided for China, which may be referable for other nations with comparable circumstances.

**Keywords:** offshore wind power; levelized cost of electricity; integrated structure design; intelligent O&M; OWP value creation

## 1. Introduction

The massive consumption of fossil fuels has brought many environmental problems and accelerated global warming, which stimulated the development of renewable energy. Wind power, as a sustainable energy source with a competitive cost, is considered to be the ideal renewable energy choice for achieving emission reduction targets. According to EMBER [1], renewable energy accounted for 38.3% of the total global power generation in 2021, of which wind power accounted for 6.7%. Without many limiting factors of onshore wind farms (e.g., noise, visual impacts, and land use), offshore wind farms are more capable of fully maximizing power generation efficiency. By 2021, the total installed power-generating capacity of global OWP reached 54.0 GW, showing that OWP has become a mainstream energy source [2,3].

In 1990, the first test offshore wind turbine (sitting on a tripod foundation) was built in Sweden (220 kW, 250 m offshore distance, and 6 m water depth). Shortly, the Vindeby wind farm in Denmark, the first commercial OWP (4.95 MW), was put into operation in 1991 with an offshore distance of 1.5 to 3 km. This meritorious wind farm, called "the cradle of the OWP industry", has provided valuable experience for the future development of OWP. In the following years, many small offshore wind farms were built in Europe, with an offshore distance up to 7 km and a water depth up to 8 m. Subsequently, offshore wind

turbines with megawatt-level capacity were developed and applied worldwide. In 2000, the first large-scale offshore wind farm, the Middelgrunden wind farm, which is composed of twenty 2 MW wind turbines, was constructed 3.5 km away from the Copenhagen Port. This project promoted the construction of another two larger offshore wind farms, namely Horns Rev I (160 MW) and Nysted (165.2 MW), in 2003. While the annual installed capacity of offshore wind farms kept breaking the record, several problems emerged. Due to the harsh marine conditions, construction costs and failure rates of wind turbines were much higher than anticipated, which limited the growth of the OWP industry for a long period of time. This is the main reason for the very slow growth rate of cumulative installed capacity during 2003 to 2006. Nevertheless, original equipment manufacturers (OEMs) and developers made great efforts to overcome many problems in the construction and operation process and finally accelerated the bloom of offshore wind farms in Europe [4]. By 2020, 162 offshore wind farms had been put into operation worldwide, and 26 offshore wind farms were under construction.

Many countries actively promote the bidding of OWP and the industrialized development of the OWP industry. It is predicted that the on-grid electricity price of OWP projects built in 2025 will likely be 0.5–0.55 CNY/kW, with a reduction of 0.3 CNY/kW from 2010 in China. The decline in wind energy price is mainly due to technological advancement, improved management, and reduction in financing costs, and the era of the connection to the grid at a competitive price is likely to come soon.

China's OWP industry has experienced nearly a decade of accelerating development (as shown in Figure 1), and the total capacity in 2021 was ranked to be the first worldwide. In 2007, China's first 1.5 MW offshore wind turbine was installed in the Bohai Sea and connected to an independent power grid for an offshore oil field. Three years later, the Donghai Bridge offshore wind farm, the first OWP demonstration project in Asia, was put into operation, which consists of thirty-four 3 MW wind turbines supplied by Sinovel were installed. From 2016 to 2020 year, the total installed capacity of OWP in China would reach 10 GW, and the grid-connected capacity would reach more than 5 GW. This also shows that it is necessary to have breakthroughs in the design and construction of a complete set of key technologies for offshore wind farms and to improve corresponding industry standards, provincial plans, and subsidy policies. From 2021, many coastal provinces in China have announced energy development plans to guide te planning, government subsidies, and technology development. For example, Guangdong province announced that the subsidies for full capacity grid-connected projects in 2022 and 2023 are 1500 CNY/kW and 1000 CNY/kW, respectively, equivalent to a one-time subsidy for the construction cost. Zhejiang province will also introduce provincial financial subsidy policies to promote the sustainable development of OWP.

Although China's OWP has developed on a massive scale, compared with favorable marine conditions in Europe, the complexity of China's marine conditions makes it more difficult to reduce the LCoE of offshore wind farm projects. Based on the data obtained from some major wind farms in China, the costs of construction and operation period are around 65–75% and 25–35% of LCoE, respectively. Figure 2 shows the cost breakdown, which prioritizes the main components, of which the costs should be reduced. Figure 3 shows the key technologies for a typical offshore wind frame, which can be categorized into the following: (1) site selection methods, (2) wind farm components, and (3) construction (O&M). In this study, Section 2 reviews the site selection methods for offshore wind farms and primary influence factors. Section 3 summarizes the key components of an offshore wind farm, including offshore wind turbines, offshore wind foundation, and booster stations. With the current solution and development trends of offshore wind farms' construction, O&M is briefly outlined in Section 4. In each section, the problems emerging in the development of China's OWP were discussed and China's efforts in technological innovation were introduced. Recommendations for the future development were also elucidated.

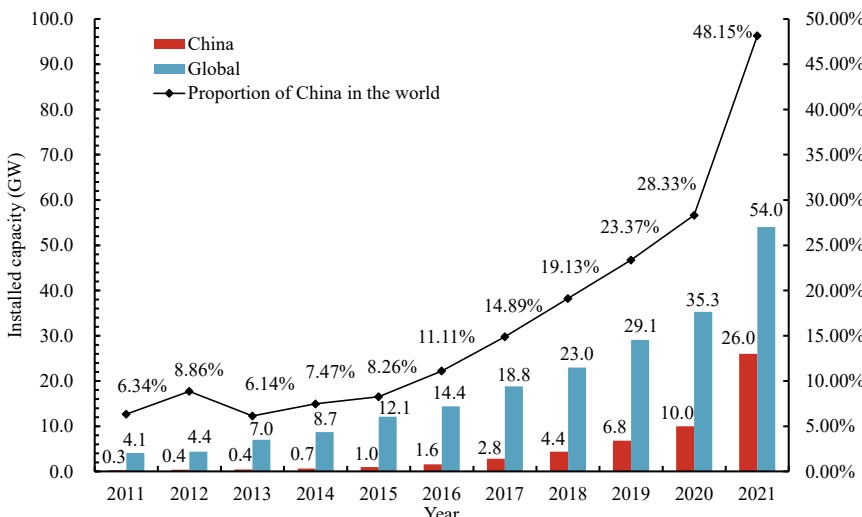

**Figure 1.** The total installed capacity of OWP over the years.

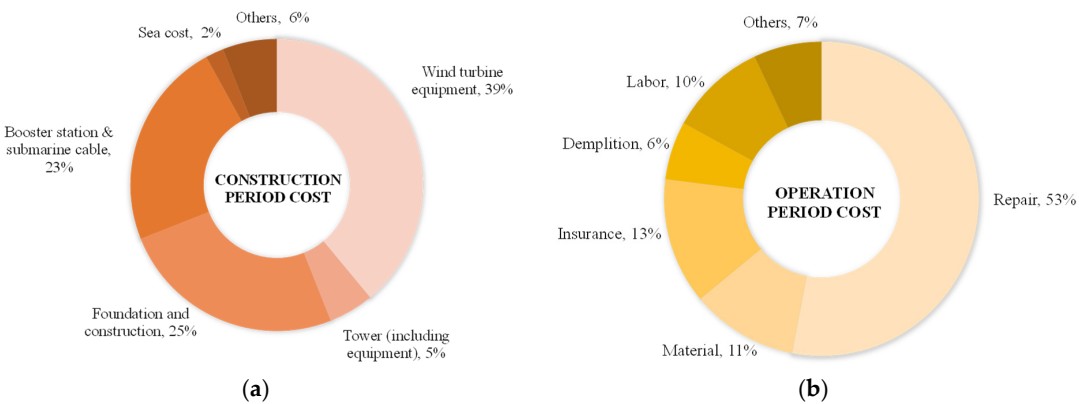

**Figure 2.** Estimated cost breakdown of China's OWP projects. (**a**) Construction period; (**b**) operation period.

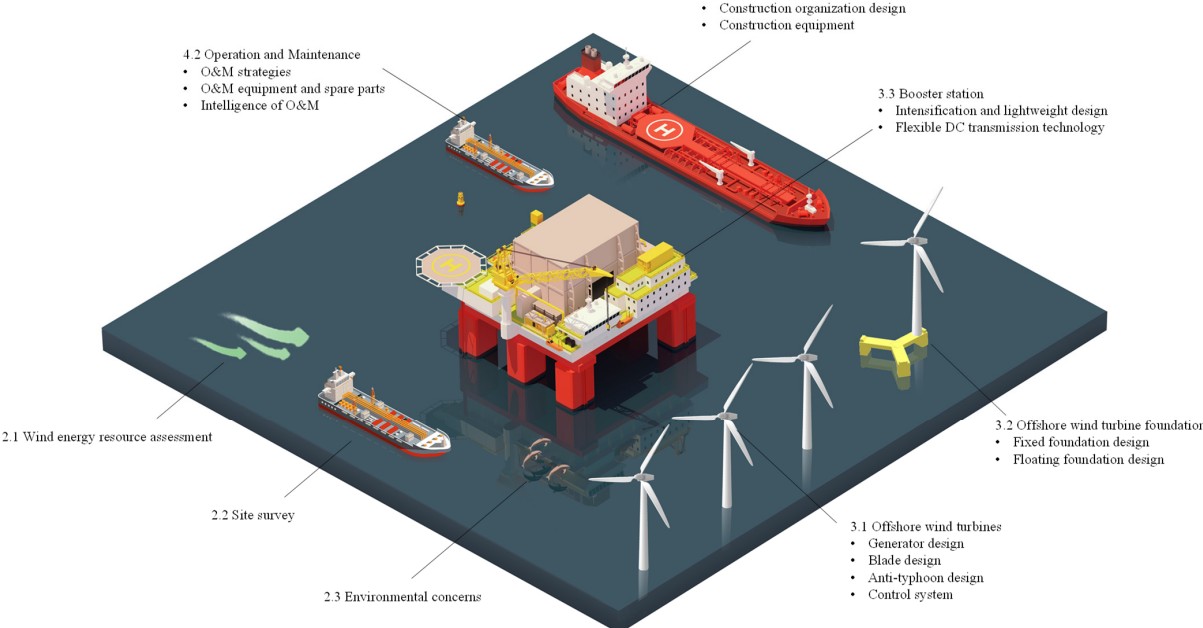

**Figure 3.** Key technologies in an offshore wind farm.

## 2. Site Selection for Offshore Wind Farm

### 2.1. Wind Energy Resource Assessment

Offshore wind energy resource assessments include macro-site selection, mesoscale simulation, and micro-site selection.

Macro-site selection is for planning the development of OWP at the national (or provincial) level. The distribution of wind energy resources over a large offshore area is first calculated based on historical weather data for resource assessment. After that, to evaluate and calculate wind energy distribution and its statistical characteristics, annual effective hours, turbulence and other variables, stake owners would carry out mesoscale numerical simulation, where wind parameters over the area under evaluation are obtained for calculating wind energy resources in conjunction with wind measurement and satellite remote sensing data. To collect sea surface wind data, wind measurement towers are built at a few representative locations. Floating LiDAR is an economical and feasible addition due to its convenient construction, reusability, and good accuracy [5,6]. In 2013, the European OWP industry began to plan the R&D pathway for LiDAR wind measurement technology; currently, many components are commercially available. Germany has clearly stated that all future offshore wind farms are suggested to use floating lidar devices for wind measurement. China is also actively developing a floating lidar wind measurement system (WindMast 350-MB). Floating lidar wind measurement is expected to replace fixed wind measurement tower in the near future. At the same time, satellite-derived sea surface wind data can be used to supplement in situ measurements. The microwave scatterometer is one of the satellite data sources for wind field monitoring at a global scale, and another widely used satellite data source is Synthetic Aperture Radar data [7]. With the monitoring data, the wind field distribution can be simulated over a large area using mesoscale atmospheric numerical models, such as MASS, TAMP, MC2, and MRF [8–10]. Mesoscale models can capture the large and medium-sized circulation process, but its parameter simplification of turbulent motion in the boundary layer is not applicable in the areas with strong turbulence (e.g., coastal area with complex terrain and active small-scale process).

Micro-site selection is the process of selecting wind turbines and optimizing the wind-farm layout for maximizing power generation. Due to the limitations of the temporal and spatial resolution and incompletion of meteorological data, high-resolution meteorological models are often used in the micro-site selection of wind farms. Currently, the commonly used software for analysis is WAsP (linear model) for relatively flat underlying surface and CFD models for complex terrain. The software for offshore wind farm site selection includes WasP, WindPro, Meteodyn WT, WindFarmer, FUGA, and Openwind. In micro-site selection, the wake effect is a critical factor for determining power generation. Katic et al. [11] proposed the PARK model, which assumes that the wake influence zone is a conical shape that is uniformly distributed along the cross-section. The wake influence zone linearly expands with the distance, while the speed attenuation mode of the wake involves linear recovery. The FUGA model proposed by Ott et al. [12] assumes that the sea surface airflow is incompressible and is a lid-driven flow, and the Monin–Obukhov similarity theory was used to describe the sea surface atmospheric boundary layer, which makes FUGA more realistic than other wake models using the single-temperature boundary layer theory. The EVM model proposed by Ainslie [13] assumes that the wake region is two-dimensional axisymmetric and uses a series of assumptions such as vortex viscous turbulence to solve the RANS equation and obtain the relevant parameters of the flow field. Some intelligent optimization methods have been successfully applied in micro-site selection. For example, a Multi-Population Genetic Algorithm program (MPGA) was used to optimize the layout of wind turbines based on the minimum investment cost and optimal power generation [14], and a genetic algorithm and a particle swarm optimizer [15] were applied to design the offshore wind farm's layout. Later, Sun et al. [16] improved the MPGA by adding a new directional restriction method to consider the influence of wind directions, which can exploit the wind resource more effectively and can be used in nonuniform wind farm design.

In the assessment of offshore wind resources, the impact of marine climate events on offshore wind farms, particularly typhoons, must be considered. Although tropical typhoons may bring a longer period of "full power" to wind farms, they typically cause serious damages to wind turbines. Therefore, it is very important to study the impact of typhoons on OWP to maximize benefits and avoid damages. To assess the impact of typhoons, historical tropical typhoon occurring within a radius of 100 km from the wind farm should be investigated for their occurring frequency, track characteristics, and seasonal and inter-annual differences. Then, the changes of wind speed and direction during the typhoon should be analyzed to provide a basis for coping with the typhoon. Extreme value analysis is conducted based on historical typhoon record to provide an extreme wind speed associated with a certain return period, which is used for selecting wind turbine.

Above all, the work content of wind resources assessment in the design includes wind resources assessment, typhoon weather, and wind turbine selection and layout (Figure 4).

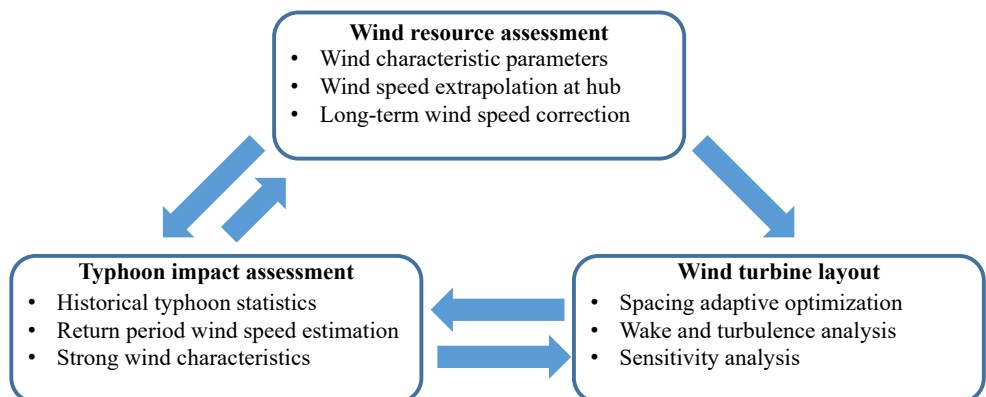

**Figure 4.** OWP resource assessment.

### 2.2. Site Survey

Site surveys provide necessary and reliable seabed topography, seabed geotechnical, and marine environmental characteristics for offshore wind farm design and construction. The survey result provides a scientific basis for foundation design, installation of offshore structures, and prevention measures of unfavorable geological phenomena.

A typical survey for offshore wind farms can be divided into four parts: (1) geophysical survey including water depth, seabed topography, seabed obstacles, and bottom profiles; (2) geological survey including the stratum spatial distribution and its physical and mechanical properties, as well as unfavorable geological and seismic factors; (3) marine surveying and mapping including surface/underwater positioning, installation positioning, and submarine cable laying positioning; (4) marine environment survey including ocean current, water level observation, water and mud temperature observation, water quality analysis, corrosion analysis, etc. The survey methods mainly include side-scan sonar, stratum profiler, multiple-beam system, magnetometer pipeline detector, high-resolution multi-channel digital seismic survey, geological borehole sampling, borehole/seabed in situ test, geotechnical test, and analysis.

In the era of OWP parity, comprehensive and detailed marine geological surveys are needed to avoid unfavorable geological conditions for initial site selection and wind turbine layout and to provide data for foundation selection and engineering design. At the same time, advanced geotechnical experiment methods are needed to obtain accurate soil conditions, combine the basic design method with soil parameters, and integrate the survey into the design, which makes it possible to increase the efficiency and, therefore, reduce the costs. In addition, under the changing marine environment, micro-site selection should also consider the changing seabed morphology. Due to the development of other coastal infrastructure, navigation, or estuarine processes, the seabed at the selected site can have some ongoing erosion/deposition, which may affect the stability of wind turbines.

Thus, these exiting morphological processes should be carefully studied and considered in the design of wind turbine foundations (e.g., propose protection measures against erosion).

### 2.3. Environmental Concerns

The construction of offshore wind farms would produce electromagnetic pollution, noise pollution, visual impact, ecological damage, etc., which negatively impact the local habitat. Therefore, the planning and site selection of offshore wind farms should follow the basic principles stipulated in national standards to meet the economic and safety requirements, as well as the needs of environmental protection to minimize the impact on birds, fisheries, and native reserves.

Some studies [17,18] have shown less bird activity in the wind farm area, which may be caused by collision with turbine blades and habitat loss. Collision mainly occurs during the bird migration activities, such as traveling between resting and foraging places or the seasonal migration of some low flying birds. In addition, most resident birds are susceptible to the noise generated by the blade sweeping and mechanical operation and avoid the offshore wind farm. Moreover, the light in the offshore wind farm is also an important factor affecting birds' safety, particularly when there is a low visibility (e.g., night time and foggy or rainy weather), since birds are easily attracted by the light. Peterson and Fox [19] found that some birds could gradually adapt to the changed habitat after the wind farm's construction. Shi et al. [20] reckons the Donghai Bridge offshore wind farm is located far away from the bird habitat and foraging area, and it almost has no influence on birds. Therefore, the layout of wind turbines could minimize the effects to birds by reasonably avoiding migration routes and primary habitats of birds and taking necessary preventive measures, such as painting warning colors at birds flying height. Meanwhile, the government also should promote the relevant environmental legislation.

For marine life, many research studies have shown that the construction of offshore wind farms mainly affects fish, benthic organisms, and plankton species [21]. While during operation, it is hard to estimate the impact, most scholars believe that electromagnetic fields and noise will affect the growth of marine life. At the same time, some studies pointed out that the underwater foundations would create new habitat for marine life, which has a positive effect on the abundance and diversity of local species [22,23]. Recently, the concept of combining marine ranch and offshore wind farm has been proposed, which uses offshore wind foundations as a place for marine life. Offshore wind farms and marine ranches can use an integrated development strategy, e.g., facility sharing and joint O&M, which could effectively reduce operational costs and ecological impact. On January, 2022, the Mingyang group completed the first harvest of fish in Guangdong Yangjiang Shaba deep-sea fishery breeding experimental area, which also marks the success of China's first "OWP + marine pasture" demonstration project.

Offshore wind-to-hydrogen is another feasible method for increasing the revenue of offshore wind farms, and China is actively looking for economically feasible methods to implement this concept. One possibility is to build onshore hydrogen facilities to absorb excessive power generation. Another idea is to produce hydrogen offshore; i.e., all power generated by wind turbines is used for production, eliminating the need for submarine cables, sea booster stations, DC/AC inverters, and transformers in the wind turbines.

### 2.4. Current Status in China

In China, offshore wind farm survey still has some shortcomings and challenges: (1) few survey ships with strong offshore operation ability; (2) incomplete survey specification and immature survey and indoor test equipment; (3) relatively low survey cost and attention. These problems lead to the severe disturbance of seabed sampling, the weak ability of advanced geotechnical tests, and the lack of comprehensive stratum information, which might bring unforeseeable risks for design, installation, and operation. This includes pile running, defined as uncontrolled rapid sinking under the weight of pile and hammer or few hammering, which often occurs in the construction of offshore wind farms with inac-

curate geological information. Therefore, China urgently needs to improve its exploration capability, build offshore exploration platforms dedicated for China's marine environment, and develop geotechnical testing equipment and technology to obtain reliable in situ test data. Meanwhile, China should improve offshore wind farm survey specifications and increase corresponding attention and investment to improve the quality of survey results.

## 3. Key Technologies of Offshore Wind Farm

### 3.1. Offshore Wind Turbines

Horizontal-axis offshore wind turbines, most used in offshore wind farms, are mainly composed of rotors, transmission systems, generators, power electronic interfaces, and grid-side step-up transformers. The rotating torque generated by the wind acting on the blades at a certain speed and angle of attack drives the blade to rotate, converting the wind energy into mechanical energy, which drives the generator to produce electricity. After being boosted by the transformer, electric energy is connected to the power grid (Figure 5). This section has summarized the development of key technology of offshore wind turbines, as shown in Figure 6.

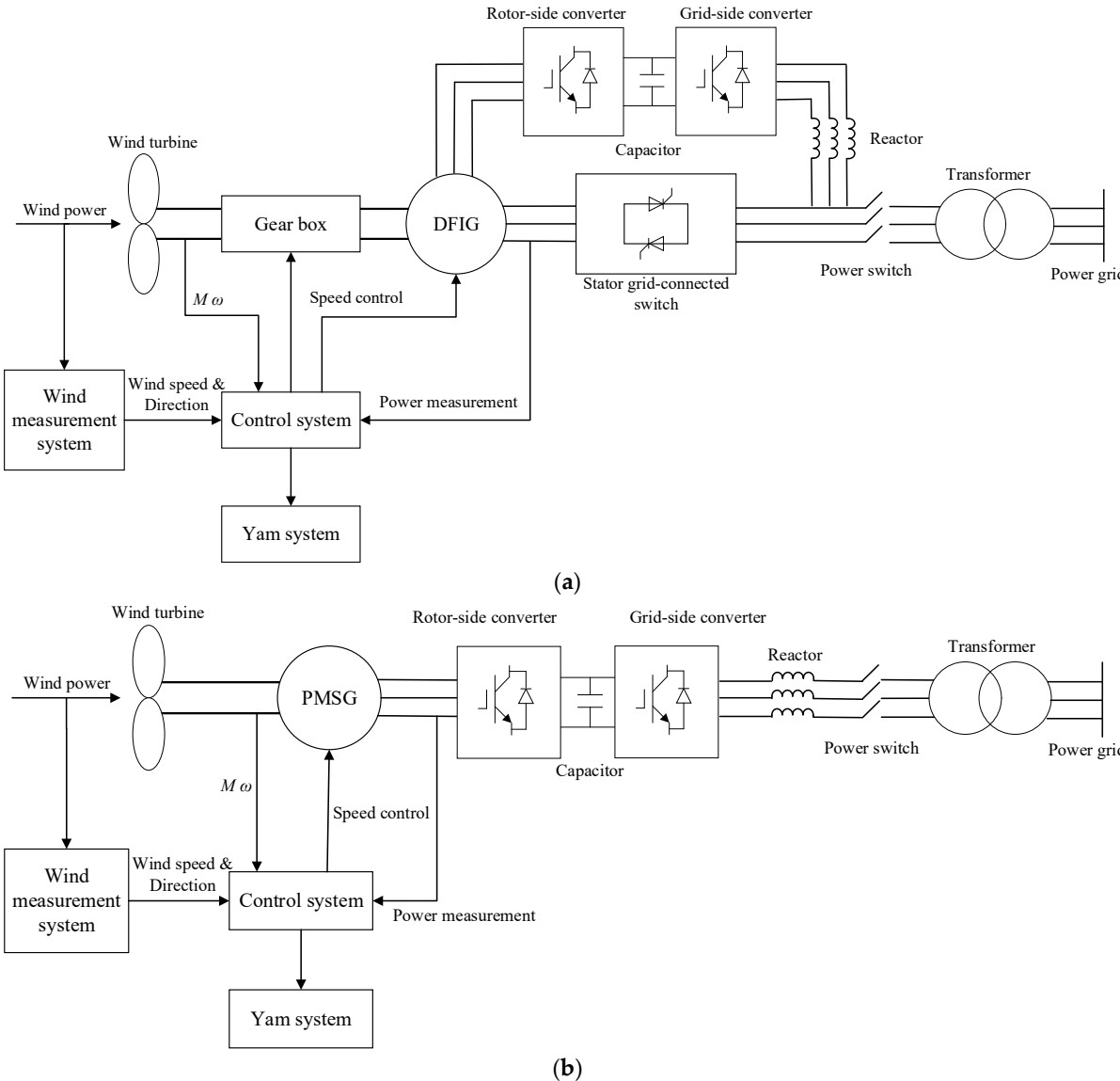

**Figure 5.** Energy flow and information flow in offshore wind turbines. (**a**) DFIG; (**b**) PMSG.

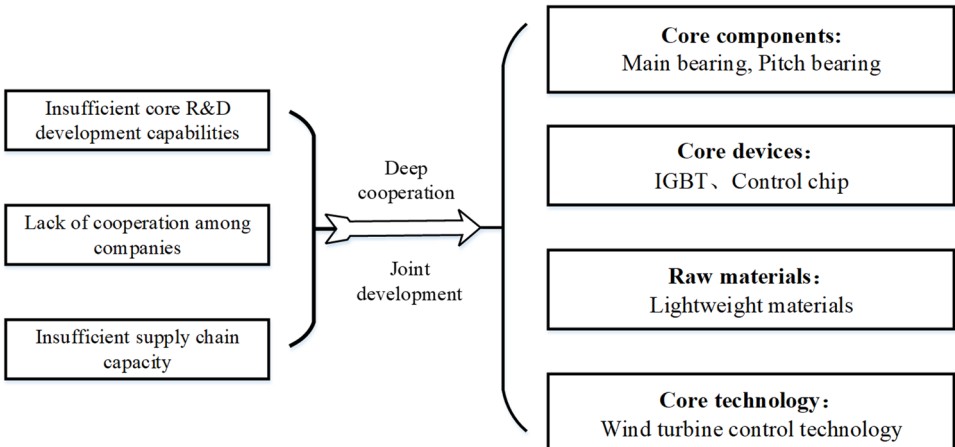

**Figure 6.** Technology development trends of offshore wind turbines.

### 3.1.1. Generator Design

The generator design of offshore wind turbines is a vital factor that directly affects the initial investment and O&M costs of an offshore wind farm. The commonly used types of wind turbine generator for offshore wind farms include the following: Double Fed Induction Generator (DFIG), Squirrel Cage Induction Generator (SCIG), and Permanent Magnet Synchronous Generator (PMSG) [24].

For a DFIG, the wind wheel is connected to the high-speed generator rotor through an accelerated gearbox, and the rotor excitation winding is connected to the grid through the rotor-side and grid-side converters while the stator winding directly to the grid. DFIG indirectly adjusts the output power of the stator side by controlling the frequency, phase, and amplitude of rotor current through an excitation converter, so DFIG has the advantages of being able to operate within a wide range of wind speed, independent adjustment of active and reactive power, and small capacity need of rotor excitation converter. In addition, DFIG is small in size and is easy to maintain, but maintenance work, such as dust cleaning and brush replacing, is frequently required, and the low voltage ride-through capacity is limited due to the small capacity of the rotor excitation converter.

SCIG is similar in structure to a doubly fed asynchronous wind turbine, while its rotor is a closed cage structure that does not need brushes and slip rings, and stator winding needs to be connected to the grid with a full-power converter. The excitation is provided by the stator-side converter, which absorbs reactive grid power. SCIG has the advantages of high reliability and wide wind speed range. However, it is not ideal for large-capacity generators, because of its low efficiency.

For PMSG, the wind wheel directly connects to the generator, and stator winding is connected to the grid through the full-power stator-side and grid-side converters. Excited by permanent magnets, there is no need to provide an external excitation power source; thus, the accelerated gearbox is not needed. However, PMSG is still large in size because of much more pole pairs required. PMSG has the advantages of high power-generation efficiency, high reliability, and strong low-voltage-ride-through capacity, but it also has the following disadvantages: large size and weight, and high cost of transportation and hoisting.

In the early offshore wind farms, SCIG was preferred due to its simple structure, mature manufacturing technology, and low cost. From 2013 to 2015 year, SCIG had been used for 75% of newly installed offshore wind turbines, while DFIG and PMSG accounted for 17% and 8%, respectively [24]. With the development of generator and blade manufacturing technology, large-capacity wind turbines are dominant. For offshore wind farms with the same capacity, large-capacity wind turbines can greatly reduce the procurement of towers, foundations, and wind turbines, as well as construction, operation, and maintenance workloads, which resulted in great reduction in LCoE; at the same time,

the application of large blades and tall towers brings a larger sweeping area and better wind energy resources, improves the utilization hours, and increases effective power generation. In the process of large-capacity wind turbines, SCIG gradually withdraws from the market due to its low power generation efficiency. In contrast, PMSG, with high energy utilization efficiency and generator efficiency, good reliability, and low maintenance costs, is more economic in full-life service and leads the OWP generator market. Many prototypes of 10 MW wind turbines are summarized in wind-turbine-models.com [25], which shows that PMSG will dominate the future large-capacity wind turbines.

### 3.1.2. Blade Design

As offshore wind turbines are becoming larger in size, the length and weight of blades will continue to increase; thus, the tip deformation and aeroelastic problems will become more prominent. This will result in a significant increase in the load on blades and the entire machine, seriously affecting the performance and fatigue of the wind turbines. Carbon fiber composite materials and balsa wood are mainly used for large offshore wind turbine blades. Compared with traditional glass fiber, these materials have the advantages of high strength, high stiffness, fatigue resistance, and light weight, which meet the requirements of long and flexible blades. The current development of ultra-long and super-flexible blades can be summarized into the following aspects:

- The aeroelasticity and bending-torsion coupling of flexible blades are investigated, and the adaptive load reduction of the blades is realized through bending–torsion coupling control, thereby reducing the blade's weight [26].
- Through the optimized arrangement of blade materials, aerodynamic damping in the blade plane is improved to achieve blade shock absorption with the smallest increase in blade weight.
- By using pneumatic accessories in conjunction with flexible blades, the range of the angle of attack for stable flow of the airfoil attached surface layer is extended, so the risk of blade stalls is reduced, and the generating capacity of the wind turbine is ensured.
- Active control technology has been introduced. By analyzing the blade's load, movement, and inflow information to obtain local strains or displacement, the controller drives the aerodynamic control system to change the coupling mechanism of the blade and the flow field, which reduces the blade's aerodynamic load.

### 3.1.3. Anti-Typhoon Design

The purpose of anti-typhoon designs is for preventing damage relative to the main structure when encountering a typhoon, of which the maximum wind speed is close to the designed speed. Extreme wind speed, abnormal turbulence, and sudden changes in wind direction are the main reasons rendering wind turbines vulnerable to typhoon attacks. Wind speed above the design condition may cause structural yield failure and a loss of control of the equipment. Abnormal turbulence could cause vibration, contributing to extreme fatigue loads. A sudden change of wind direction would easily cause damage to the yaw system. Anti-typhoon design needs to consider the failure loss and failure mechanism; set appropriate safety factors for the wind turbine, tower, and foundation; and even allow the blade to yield when it is necessary to reduce the loads acting on the entire structure and to protect the tower and foundation. In operation, reasonable control strategies of braking, yaw, and pitch should be adopted to minimize damages.

### 3.1.4. Control System

The control system mainly includes the automatic yaw system and pitch control system, and its overall goal is to maximize power generation, reduce the load on mechanical components, and protect the wind turbine when the wind speed is too high. The yaw system can quickly and smoothly make adjustments when the wind direction changes so that the wind turbine can efficiently harvest wind energy. Pitch control system is a

device installed in the hub as air braking or power control of wind turbine operation by changing blade angle. When the wind direction changes, the yaw system will send the yaw command through the output signal of the wind direction sensor and control the yaw motor to drive the yaw gearbox to perform the correction; thus, the wind wheel can harvest the maximum wind energy. In recent years, scholars have carried out many studies on the active yaw control algorithms for large wind turbines, including Hill Climbing algorithm, Vane–Hill Climbing algorithm, Kalman–Hill Climbing algorithm and Fuzzy control algorithm, etc. [27,28].

The pitch control system enhances the capture of wind energy, and optimizes the aerodynamic torque and aerodynamic force of the wind turbine by controlling the pitch angle through the bearing installed in the hub. When the wind turbine operates below the rated wind speed, the pitch control system sets the pitch angle to zero to render the wind wheel in the maximum capture state. When the wind speed exceeds the rated wind speed, it will adjust the pitch angle to stabilize the generator's output around the rated output. Pitch control is divided into unified blade pitch control and individual blade pitch control. The former makes each blade change the same angle to realize the adjustment of blade speed and output power; the latter controls each blade's pitch angle individually to balance the load and reduce the oscillation. Some advanced wind turbines use the combination of the two. The output power is adjusted through a unified pitch control, and the load of each blade is optimized by using independent pitch control to enhance the stability of the wind turbine [29,30]. The Proportional Integral Derivative (PID) control method is widely used in pitch control, while large-capacity wind turbine is a typical non-linear, multivariable, strong coupling, and time-varying system. However, PID control may not meet the requirements of control accuracy in such complex marine conditions. Therefore, the development of pitch control focuses on intelligent control methods such as fuzzy control, neural network control, etc. [31,32]. With respect to pitch power, there are electric and hydraulic pitch systems. Compared to the electric pitch system, the hydraulic pitch system has the advantages of small volume, light weight, good dynamic response, large torque, and no need for a speed change mechanism. At this stage, the major manufacturers, such as Vestas and Siemens, all adopt the hydraulic pitch technology, which accounts for more than 70% of the offshore installed capacity.

### 3.2. Offshore Wind Turbine Foundation

The foundation plays an essential role in ensuring the stability of the wind turbine. There are two general types of foundation: fixed and floating foundations. The former is suitable for shallow waters, and it includes monopile foundation, gravity foundation, suction bucket, tripod foundation and jacket foundation, etc. (Table 1). Most existing offshore wind turbines have fixed foundations, especially monopile foundations [33]. As water depth increases, the cost of fixed foundations increases sharply; thus, floating foundations become more suitable choices for deep waters. However, commercial-ready applications still require research on the overall structure dynamic characteristics, the foundation bearing mechanism, soil failure mechanism, the structure design method, and the construction technology.

### 3.2.1. Fixed Foundation Design

The foundation design mainly carries out limit state and fatigue analyses for the foundation structure under the loads including wind, wave, current, ice, earthquake, and superstructure load, considering the scour [34] and dynamic responses of the structure [35] as well (the natural frequency of the structure must not match the external load frequency) so as to provide a basis for the selection of foundation structure. There are two general methods for designing foundation: the integrated design method and SIA (Sequentially Iterative Approach). The integrated design method takes the wind turbine, tower, foundation, and external environment as an overall system, and it obtains accurate structural dynamic response through integrated numerical analysis. DNV.GL proposed the integrated design

concept of the offshore wind turbine foundation (Project FORCE) in 2014, which can reduce LCoE by 12%. This concept includes overall design and relaxation of frequency constraints. The integrated design enables better connection between wind turbine manufacturing and support structure design, improving design efficiency and accuracy; at the same time, it relaxes the design range of structure natural frequency; thus, a structural design with higher rigidity is allowed, greatly saving the amount of steel used.

**Table 1.** Comparison of several fixed foundation types.

| Foundation | Technical Principle | Water Depth | Advantages | Disadvantages | Applications |
|---|---|---|---|---|---|
| Monopile | Hollow steel cylinder inserted into the seabed to provide resistance | <30 m | Light weight, simple structure, convenient installation, no need to tidy up the seabed | Largely restricted by geology, need anti-scouring device | Anholt, London Array, Dantysk |
| Gravity | Mass concrete stabilized through its own weight | <30 m | Simple structure, low cost, high stability, and reliability | Need to be pre-treated, long construction period, large and heavy foundation, inconvenient T&I | Thornton Bank phase I |
| Tripod | Three diagonal braces and horizontal braces welded and connected to the steel pile, forming the frame to bear and transmit loads | 20~80 m | Light weight, good stability, suitable for hard seabed, anti-scouring. | Restricted by the geological conditions, need a fixed offshore construction platform | Alpha Ventus, Trianel Windpark, Borkum I, Global Tech I |
| Jacket | Composed of grid row trusses, three or four steel pipes are driven into the bottom to fix the foundation | 10~80 m | Light weight, high strength, low requirements for piling equipment. | High steel demand, construction is severely affected by weather | Thornton Bank phase II, III |
| Suction bucket | The negative pressure between the bottom of drum and soil keeps the whole system stable | 5~60 m | Quick installation, no need for large-scale T&I, and seabed pretreatment | Easy to tilt, need frequent correction, difficult to design | Frederikshavn, Qidong, Xiangshui |

Since tower and foundation are usually supplied by two unrelated manufacturers, it may not be possible to adopt the integrated design method when the parties involved do not want to share confidential business information. Under such circumstances, SIA is the only choice. SIA takes the tower bottom as the interface; thus, the participants carry out design and optimization iterations for the tower and foundation, respectively. The wind turbine manufacturer usually carries out modeling and the load calculation of the upper support structure first; then, it checks the tower structure for the limit and fatigue load of each interface of the tower and then passes the design of the upper part to the foundation to the foundation designer for further designing. In this process, wind turbine manufacturers and foundation designer have repeatedly considered the effects of loads such as waves and currents. Manufacturers and design institutes search for optimal local designs in their respective design domains, causing conservative structural design.

### 3.2.2. Floating Foundation Design

By the end of 2020, the total capacity of floating wind turbines or commercial projects in operation in the world was 125 MW. The Global Wind Energy Council predicts that

6.2 GW floating wind power projects will be built in the next ten years according to the existing floating wind power planning of major countries worldwide.

Reducing floating wind turbine's cost will primarily come from the design optimization and improved manufacturing process of the wind turbine and floating platform. Future research should target improving our understanding and modeling capability of the loading characteristics, overall bearing capacity, and system's dynamic response. Secondly, the mooring system is one of the major technical bottlenecks in designing offshore floating structure and is the key to cost control. Common types of mooring include catenary mooring structure and tension line. For the catenary mooring system, the restoring force of the anchor chain is mainly realized by the change of the suspended section. It has good strength and simple manufacturing process, but the weight (cost) increases dramatically with the depth of application for needing long lay-flat sections on the seabed. Therefore, tension lines (steel cables) are often used for mooring in deep water, and the binding force of the mooring lines mainly depends on the tensile deformation of the steel cables. The tension line uses less steel, but its anchorage section has to bear both horizontal and vertical forces. Meanwhile, the mooring line stiffness also has large non-linear characteristics such as relaxation and creep, which increases the difficulty and cost for maintenance. In addition, the dynamic cable is another high-risk source of floating wind power operation, and its mechanical and electrical performance also requires integrated analysis with the wind turbine, floating platform, and mooring structure.

Several main types of floating foundations have been developed and reached the stage of prototype test:

1. Spar: Spar foundation is a closed cylindrical buoy anchored to the seabed by mooring lines. Its center of gravity is far lower than the center of buoyance, making the cylindrical buoy float upright. The small waterplane design can reduce the heave motion of the platform. Hywind Scotland pilot park, the first floating offshore wind farm, adopted this type of foundation. With an installation depth of 100–120 m, the wind farm is equipped with five sets of 6 MW wind turbines and is connected to the grid in the UK in 2017.
2. Tension leg platform (TLP): The TLP idea comes from the offshore oil platform. The system is partially submerged below the sea surface and anchored to the seabed via pull rod or cable. The buoyancy force on the submerged part to support the wind turbine and maintain the tension of the pull rod or cable. TLP can have very little vertical movement, but its installation process is quite complex, and the applicable water depth is usually more than 40 m. In 2012, SWAY deployed a single-tension leg floating offshore wind turbine in Norway.
3. Semi-submersible: The semi-submersible foundation is formed by connecting a few cylindrical pontoons in a triangular or rectangular layout, which provides buoyance force. Catenary mooring lines connect the floater to the seabed. The applicable water depth is usually more than 40 m. Principle Power, a developer of floating offshore wind turbines, installed a semi-submersible foundation named wind float on the Portuguese coast in 2011, with a capacity of 2 MW.
4. Barge: The barge-type foundation platform is like a vessel. The water depth is usually more than 30 m. The barge has the advantages of simplicity, easy manufacturing, and good stability. It can be towed to any location, allowing flexible deployment and low cost. Floatgen, France's first floating wind turbine test project, uses a concrete barge-type floating wind turbine foundation with a damping pool. One 2 MW prototype was deployed, and semi-tensioned mooring is realized by six polyester mooring lines.

*3.3. Booster Station*

3.3.1. Intensification and Lightweight of Substation

The offshore booster station collects all the power collection lines and then boosts and transmits power. It also serves as the control center of the offshore wind farm. With the increasing capacity demand of offshore booster station, the construction cost has

also risen sharply with the increasing weight of the superstructure. The lightweight substation and its intensification are suggested for reducing the construction period and cost. Recently, CIMC Raffles and Siemens Energy jointly developed 300 MW and 500 MW prefabricated modular offshore substation (PMOS) and obtained the approval in principle (AIP) of the China Classification Society (CCS) in 2021. The electrical equipment in PMOS adopts a prefabricated cabin for "modular" configuration, for which its overall dimensions and fixing methods adopt the existing industrial product standards. The installation, commissioning, and test of various functional equipment in the prefabricated cabin can be completed in the factory. For example, pluggable cable terminal connections are used between electrical modules, which can realize electrical connections inside the substation without on-site construction and, at the same time, avoid potential safety hazards caused by cross-construction. Based on the design experience of more than 50 offshore booster stations, the Huadong Engineering Corporation released SLIM-SO offshore booster station products, including integration installation solutions and modular installation solutions. Compared with the booster station of similar on the market, this series of products reduced the projection area by 25~35%, the weight per MW by 30~40%, the cost by 15~25%, and the production cycle by 20~30%.

### 3.3.2. Flexible DC Transmission Technology

After decades of development, high voltage direct current (HVDC) technology is becoming mature. Among them, flexible DC transmission technology based on VSC or MMC and IGBT has great technical and economic advantages for OWP projects with an installed capacity over 500 MW. Flexible HVDC technology has the following advantages [36–39]: (1) Flexible HVDC converter station can continuously and independently control active power and reactive power, and the operation control mode is flexible; (2) it has good fault ride-through capability and black start capability and small space requirements for reactive power compensation device and filtering equipment. China has made significant progress in the R&D and manufacture of key equipment for flexible DC transmission, including high power electronic devices, converter valves, HVDC circuit breakers, HVDC converter transformers, and DC cable, etc. [40] For example, in 2017, China Railway Rolling Stock Corporation (CRRC) developed 4500 V/3600 A press-pack insulated-gate bipolar transistor (IGBT), which achieved a breakthrough in China's relevant technologies. Subsequently, it developed 4500 V/4000 A reverse-conduct IGCT products. In recent years, the voltage level and transmission capacity of China's flexible DC transmission projects have reached the world-class level. On 25 December 2021, the Rudong Jiangsu OWP Flexible DC Transmission Project realized full-capacity grid connection, becoming the first OWP flexible DC transmission project in Asia.

### 3.4. Current Status in China

Insufficient supply chains have always been a hindrance to the leapfrog development of China's OWP industry; thus, achieving the localization of the manufacture of key components of offshore wind turbine is the primary target of China's offshore wind industry in the past decades. As the Chinese government cancels the subsidy policy for OWP projects after 2021, a rapid installation period has swept through China's OWP industry, leading to an imbalance in between the supply and demand of key components' production capacity, especially for the main bearings and blades of wind turbine. At the same time, affected by the suspension of international logistics, shortage of raw materials, and delay in the resumption of labor resources caused by COVID-19, the constructions of offshore wind projects were generally delayed for one to three months. However, such a severe situation has also directly promoted the resumption in all parts of the industrial chain and the adjustment of local industrial structure. During the epidemic period, the local government, developers, and manufacturers jointly continued to develop OWP industry bases in Dafeng, Yangjiang and Fuqing, which enabled China in producing 10 MW wind turbines, large transformers, and submarine cables of various voltage levels. Despite the challenges, China

still maintains the world's largest new installed capacity, demonstrating the strong flexibility and resilience of China's OWP industry. However, there is still a gap with advanced foreign enterprises in the fields of the high-performance bearings. For example, the manufacture of large PMSG generators, which are more popular for wind turbine above 10 MW, is restricted by the manufacturer's existing manufacturing capability, and it is difficult to ensure the processing accuracy and production quality of key components. For another crucial component, IGBT, China has obtained the key technology and the entire process of high voltage IGBT manufacturing and initially built a complete industrial chain of high voltage IGBT. However, the high-performance IGBT used in the high-voltage flexible DC converter of OWP still relies on imports, and the main suppliers are ABB, Infineon, Toshiba, Mitsubishi, etc. These suppliers own the key technology of the flexible DC transmission system's large-capacity converter of OWP. It has been applied in large-scale offshore wind farms abroad and accumulated many application experiences. There are mature research results in China's main circuit topology, control and protection strategy, modeling and simulation analysis, transient voltage stability, and design and manufacture of high voltage and large capacity flexible DC converter. Still, there is little OWP engineering application experience, and it needs to be further extended.

For offshore wind turbine design, China's research in wind turbines is relatively insufficient. For instance, most of the software used in wind turbine design and load evaluation are products of European companies. The design standards and concepts are in accordance with the requirements of wind turbine certification rules proposed by DNV and wind turbine technical standards proposed by the International Electrotechnical Commission (IEC), without fully considering the particularity of China's wind energy resources, natural environment, and grid acceptance mode. China's OWP industry still needs to motivate the "integrated design". Therefore, domestic owners should actively promote the integrated design cooperation between the complete machine manufacturer and the design institute and promote the unification of design standards, the integration of overall modeling, working conditions, and the overall dynamic loading integration process. Some leading domestic wind turbine manufacturers, such as Goldwind, and design institutes have cooperated in carrying out integrated design work and proposed an integrated design method, iDO (Integrated Design Optimization), based on a digital cloud platform. The calculation results show the following: under ULS (Ultimate Limit State) and FLS (Fatigue limit state) conditions, the structural strength, deformation, fatigue damage, and other indicators of the integrated design method based on the iDO cloud platform are significantly lower than those of the distributed iterative approach (SIA), which can significantly optimize the tower foundation structure, reduce the structural weight, reduce the cost of the entire support structure, and reduce the LCoE of OWP [41].

## 4. Key Technology of Construction, Operation, and Maintenance

Marine operation is limited by the sea conditions. The most significant influencing factors are strong wind and waves [42]. For example, the maintenance and repair work of malfunctioned wind turbine are (stipulated by Standardization Administration of China) to be carried out under the conditions of wind speed less than 12 m/s, but the actual conditions on the sea often exceed such requirements, resulting in extending the outage time. Therefore, a good use of the limited window period with acceptable sea condition is important to reduce construction and O&M risk, improve efficiency, and save costs.

### 4.1. Construction

4.1.1. Construction Organization Design

The construction of offshore wind turbines and foundations has the following requirements [43]: (1) good transportation and hoisting capability to cater for large and heavy wind turbines and foundations; (2) high efficiency requirements to cope with the limited window period; (3) good installation accuracy. Since a small problem will possibly cause extensive delay and additional cost, the developer needs to optimize the construction organization.

The construction of offshore foundation has been relatively mature; thus, this section mainly reviews the construction technology of wind turbine and tower. The construction of offshore wind turbines has two main approaches, i.e., overall transport and installation (T & I) and separate T & I. For overall T & I, the entire wind turbine is assembled on shore, and then it is transported to the site for installation onto the foundation. In order to ensure stability during transportation, a hoop apparatus is generally used to fix the wind turbine at the middle section of the tower. In addition, it is necessary to pay attention to avoid collision between wind turbine tower and platform and ensure the alignment accuracy;l thus, the stability check of the installation ship is required. The hoisting is not limited by the water depth, which can reduce the uncertainty in the installation process and reduce the standby time of ships and personnel, thereby reducing project implementation costs and risks. However, it has high requirements for installation ship, hoisting slings, and transportation fixtures. The thirty-four 3 MW wind turbines of the Donghai Bridge Offshore Wind Farm, the first large-scale offshore wind farm in China, adopted this method for transportation and installation. Separate T & I components are transported separately and assembled onsite, which reduces operating time and offshore installation difficulty. There are two main assembly schemes: One is to assemble three blades and hub into a wind wheel system for transportation, and the other is to onshore assemble two blades, hub, and nacelle for transportation, similarly to "rabbit ears". The offshore T & I equipment requirements for separate T & I are relatively low. However, the installation process can be significantly affected by the marine environment, meteorology and geological conditions because of its high workload, so high stability ships such as the jack-up platform are usually used.

4.1.2. Construction Equipment

Engineering ships for OWP mainly include crane vessel, submersible vessel, non-self-propelled jack-up vessel, self-propelled jack-up vessel, etc. The self-propelled jack-up vessel has a large deck equipped with special cranes and piling equipment, which can carry multiple wind turbines as well. Meanwhile, it has high speed and good maneuverability, and the widest scope of application. The requirements for the hoisting height and capacity, operation depth, and specialization of ships are becoming higher, and the construction ship resources suitable for the next generation of wind turbines, especially for wind turbines above 10 MW, are facing a shortage. China's OWP industry is actively addressing this current shortage. In 2022, the 3000-ton Wudongde, a crane vessel, and the 2000-ton Baihetan, a self-propelled jack-up vessel, were jointly built by enterprises such as China Three Gorges Corporation, which can meet the construction requirements of 10 MW+ offshore wind turbines and ex-tend the annual window by 30 days.

*4.2. Operation and Maintenance (O&M)*

4.2.1. O&M Strategies

Existing offshore wind farms' O&M strategies are mainly divided into two categories, namely preventive maintenance (PM) and corrective maintenance (CM).

Preventive maintenance refers to the maintenance before component failures to keep the wind turbine in regular operation, which is a proactive maintenance strategy. According to different maintenance concepts, preventive maintenance can be divided into time-based maintenance (TBM) and condition-based maintenance (CBM).

Time-based maintenance is also known as periodic maintenance, which makes a maintenance plan for each component for periodic maintenance based on equipment failure patterns. The periodic maintenance cycle is usually half a year, one year, or five years [44]. An unreasonable formulation of regular maintenance plans will lead to excessive maintenance or insufficient maintenance; at the same time, due to the harsh marine environment, the aging law of wind turbine components may deviate from the theory, and sudden failures may occur. Therefore, it is not enough to adopt regular maintenance only. Hofmann and Sperstad [45] considered time-based preventive maintenance in their maintenance model, defining fixed maintenance intervals for components and maintaining their state

to the initial value. Sahnoun et al. [46] considered a systematic preventive maintenance strategy based on a determined timetable, which is most effective when components are periodically degraded.

Condition-based maintenance, also called predictive maintenance, comrpises arranging a maintenance plan to prevent further component deterioration when specific condition monitoring signals are abnormal. Hinrichs [47] compared the cost of applying condition-based and post-maintenance strategies, and the results show that the cost of condition-based maintenance is lower. Therefore, he suggests that once the monitoring signal of the component shows the trend of damage, it should be maintained in the next routine maintenance. Fu and Yuan [29] reviewed and classified the existing research on condition monitoring technology from condition information collection, condition information transmission, fault diagnosis, wind turbine condition control, operation cost analysis, monitoring system development, etc.

Reliability-centered maintenance strategy and opportunistic maintenance strategy are two methods belonging to CBM. Reliability-centered maintenance (RCM) is a comprehensive method of adopting passive, preventive, and active maintenance strategies in order to increase the probability of normal operation of machines or components in their design life cycle with the least amount of maintenance. RCM takes the expert system and engineering experience as the decision-making basis, carries out the failure mode research and effect analysis (FEMA) for main components, and makes maintenance plans after clarifying maintenance requirements. Igba et al. [48] discussed applying RCM to the gearbox maintenance of wind turbines from the perspective of system approach and proposed a systematic framework of using RCM to the maintenance of wind turbines. Pattison et al. [49] proposed an RCM architecture for the offshore wind turbine, including intelligent condition monitoring, reliability, and maintenance modeling, and maintenance schedules to maximize the availability and profitability of the wind turbine. Du et al. [50] proposed an RCM framework suitable for offshore wind farms and pointed out that the wind turbine fault monitoring technology based on data of supervisory control and data acquisition system (SCADA) and the management technology of spare parts and transportation tools are the key technologies under this framework.

Opportunistic Maintenance Strategy comprises making full use of the opportunity of meteorological forecast (mainly wind condition forecast) and corrective maintenance to carry out preventive maintenance at a low cost. Specifically, preventive maintenance is carried out when the wind speed is small or corrective maintenance is needed to reduce the shutdown loss and transportation cost, thus reducing overall maintenance costs. Besnard et al. [44,51] has put forward an optimization framework and stochastic optimization model of opportunistic maintenance strategy for the offshore wind farm, and the results show that opportunistic maintenance strategy can significantly reduce maintenance costs.

Corrective maintenance, a passive maintenance strategy, is also called post-maintenance. Onshore wind farms usually adopt this maintenance strategy because of their good accessibility. Once there is a failure, it can be repaired in time. The early maintenance of offshore wind farms is directly borrowed from the onshore post-maintenance strategy, including routine inspection, corrective maintenance and regular maintenance. However, offshore wind turbines may not be repaired in time due to the harsher environment, resulting in excessive downtime and losses. In addition, post-maintenance often requires more maintenance resources, increasing maintenance costs [52]. At the same time, due to the lack of understanding of components' failure characteristics, this maintenance strategy cannot formulate targeted maintenance plan for specific needs, resulting in insufficient or excessive maintenance.

### 4.2.2. O&M Equipment and Spare Parts

Due to the unique environment of offshore wind farms, the O&M work needs to rely on professional O&M ships. At present, the O&M ships used in China are mainly ordinary O&M ships developed from transportation boats and fishing boats, most of which are

steel monohulls, which cannot meet some users' needs. With the offshore wind farms gradually moving towards far and deep waters, O&M ships face challenges such as greater offshore distance and worse sea conditions, where ordinary O&M ships will not meet the requirements. Therefore, there is an urgent need for professional O&M ships that can match the maintenance loading and hoisting capacity of large wind turbines. Meanwhile, as the supply chain of offshore wind turbines is not mature enough, a possible shortage of spare parts required for maintenance could occur during the limited offshore operation window, resulting in extensive wind turbine outage losses. Therefore, reasonable spare parts management is necessary. Currently, the spare parts management modes of offshore wind farms mainly include spare parts inventory established by manufacturers, owner-built spare parts inventory, and third-party centralized inventory [53].

When the difficulty of O&M becomes greater with increasing offshore distance, the shutdown of wind turbines would bring more power loss. Helicopters, with higher safety and commuting efficiency, are beginning to be used in the operation and maintenance of several offshore wind farms in Europe. By taking an offshore wind farm 60 km offshore, it takes 5 h to arrive by a vessel, while only 30 min is needed to take a helicopter. Obviously, it is also more useful when there are casualties at sea. On the other hand, the wave height and wind speed limit are lower for helicopters. Silva and Estanqueiro [54] analyzed the accessibility of three offshore Portuguese maritime regions considering boat with OAS and helicopter, and they found that boats with OAS technology have an availability between 80 and 90% in the summer period and 60% in the winter season, while it is nearly 100% for helicopter throughout the year. As the number of helicopter used for maintenance is increasing, Strbac et al. [55] examined the sizes of the flight corridors on offshore wind farms and the lateral safety clearance for helicopter hoist operations at offshore wind turbines. On the other hand, there are still no cases of adopting helicopters for the maintenance of offshore wind farms in China.

### 4.2.3. Intelligence in O&M of Offshore Wind Farm

Several problems still exist in current offshore O&M, such as poor accessibility, incomplete condition monitoring and fault diagnosis functions, few professional O&M personnel, and a shortage of O&M resources. Thus, it is necessary to manage the O&M plan and time intelligently, and the development trend of operation and maintenance of offshore wind farms is shown in Figure 7. Topics of research focus with respect to intelligent O&M technology are drawn as following:

- Integrated information intelligent management: integrate information of wind turbine, personnel, vessels management, meteorology, O&M tools, and spare parts management, which supports the planning of various O&M tasks.
- Intelligent condition monitoring and fault diagnosis: Based on the intelligent condition monitoring technology, intelligent online diagnosis, and prediction of faults, the hidden dangers of wind turbines can be discovered and maintained in time, which could greatly improve the work efficiency of the wind farm and reduce the O&M cost.
- Intelligent monitoring and operation optimization: Utilize data from various sensors, such as wind farm meteorological data, grid power demand, wind turbine health status, etc., to adjust the operation status of wind turbines in order to improve the utilization rate of wind energy.

The safe operation of offshore wind turbines relies on sensors to continuously provide reliable information such as turbine vibration and temperature of key components to diagnose the operating status of the wind turbine. At present, typical sensors used in offshore wind turbines include wind, vibration, vibration switch, yaw position measurement, speed, temperature and humidity, main control electric energy meter, lidar wind measurement, tower clearance monitoring, strain, lightning current, inclination, etc.

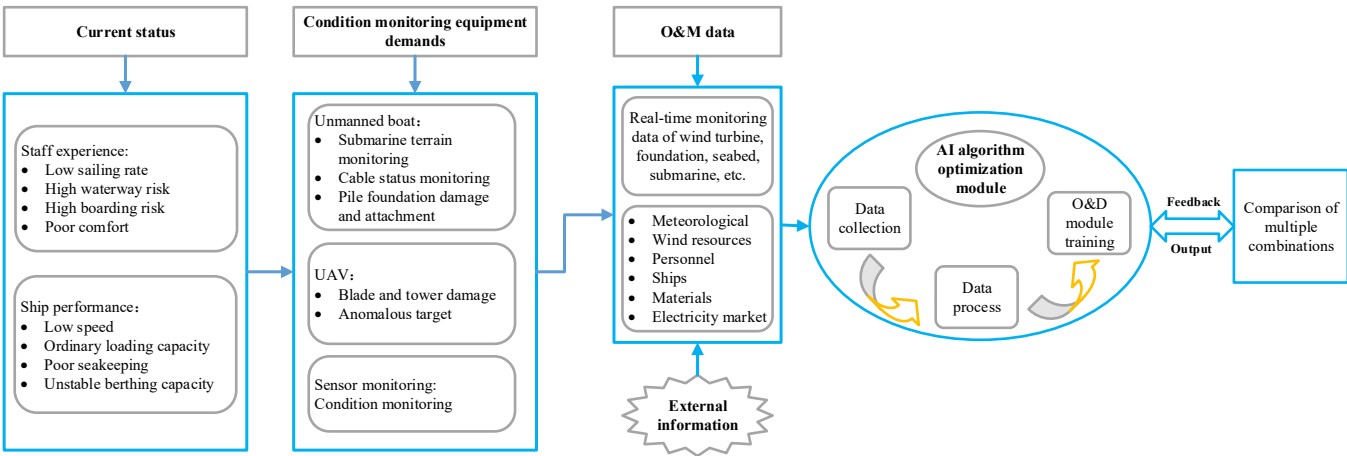

**Figure 7.** Development trend of operation and maintenance of offshore wind farms.

### 4.3. Current Status in China

China has accumulated a lot of construction experience in large-scale OWP construction and has formed mature construction procedures, especially for the construction of high-pile cap foundations. However, some key construction technologies still need to be developed for more complex marine construction environment, such as pile running controlling, underwater grouting technology, and anti-scouring technology. Currently, research on offshore wind farm construction is mainly based on specific project practices, with a primary focus on construction scheme technology [43,56,57]. These studies suggested that the construction concept of an offshore wind farm should be adapted to the marine construction environment. Hydrogeological and meteorological information must be used to guide the selection of marine equipment. The construction plan should be optimized by accounting for total equipment situation and effective window period. Improving the installation efficiency is the main trend in the development of offshore wind farm construction (Figure 8). The average installation time of a single wind turbine is expected to be shortened to within 2 days through integrated T&I, hoisting process optimization, construction resource scheduling optimization, meteorological monitoring, etc.

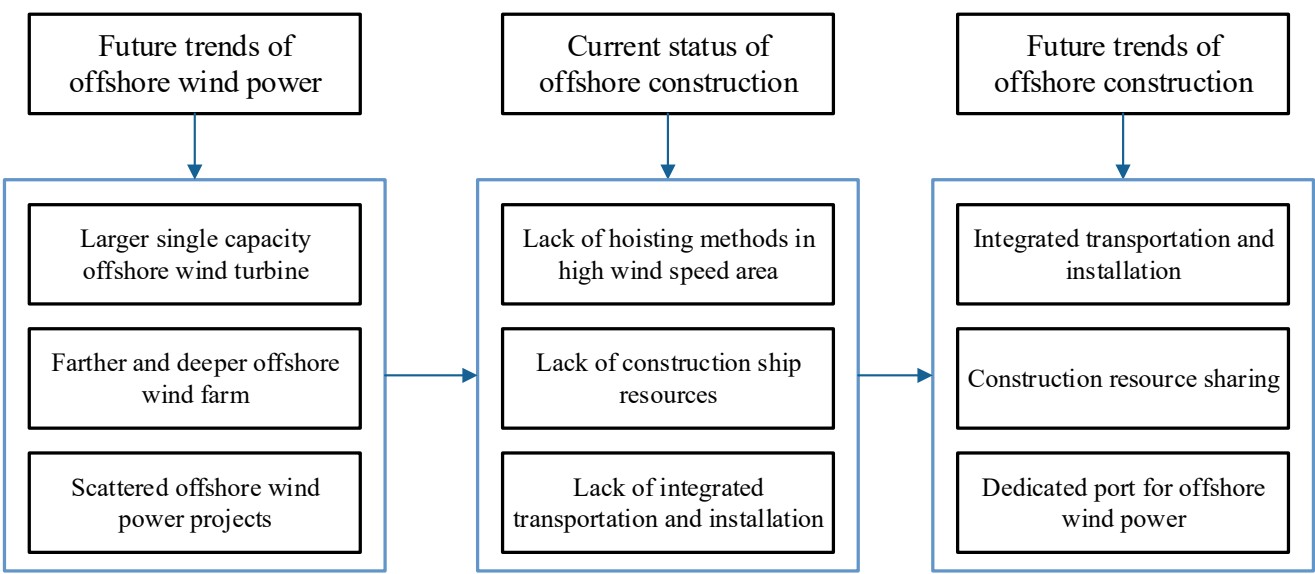

**Figure 8.** Development trend of offshore wind farm construction.

For offshore wind farm maintenance, China still adopts manual and regular inspections, which have problems such as long periods, high costs, and high-security risks as

offshore wind farms are far from the shore and have poor accessibility. Thus, uncrewed intelligent inspection equipment and method for detecting wind turbines, infrastructure, submarine cables, and sea conditions is currently a research hotspot for key operation and maintenance equipment. At home and abroad, the developments of advanced equipment are promoted, such as regional surveillance UAV (Unmanned Aerial Vehicle), large-scale uncrewed boat, and the underwater robot, mainly focusing on the key technologies such as accurate positioning and attitude control, image recognition, and remote communication of equipment in the complex marine environment so as to realize remote terrain change analysis, cable condition monitoring analysis, pile foundation damage analysis, blade, tower damage detection, sea surface abnormal target monitoring, etc. Currently, leading operators and manufacturers in China are actively engaged in researching and applying O&M management technology. Goldwind has developed iGO offshore wind farm intelligent management system, which can establish files for each wind turbine and manage the full life cycle information. By using machine learning, neural network, and other methods, the failure models of key components were established to realize fault diagnosis and transform from failure maintenance to early warning maintenance, and this reduces O&M costs by 15~20%.

Chinese businesses have kept developing new technologies in the sensor industry. There are domestic manufacturers of the frequently used sensor types specified in Section 4.2.3. For some specific scenarios with higher requirements for the accuracy and reliability of the sensor itself, manufacturers will prefer to use imported brands to assure the accuracy and stability of data collection. It can be seen that domestically produced middle- and low-end sensors for wind turbines have no technical issues and are quite competitive on the market, but they are high-end sensors that are lagging behind in terms of technology. In light of the development trend of wind turbine sensors, it is advised to conduct research and development on customized and domestic sensors, edge computing, the Internet of Things (IoT), and intelligent monitoring platforms. After that, domestic high-precision new MEMS (Microelectro Mechanical Systems) sensors, sensor edge computing algorithms and equipment, and finally a wind turbine intelligent sensing condition monitoring platform can be developed.

## 5. Research and Development of OWP in China

### 5.1. China's R&D Efforts in Offshore Wind Industry

From 2010 to 2020, the SCI database collected 7573 articles in OWP (search term: offshore wind), including 1485 articles published by Chinese scholars. Figure 9 shows that the articles by Chinese scholars both increased in quantity and percentage, at a peak of 391 articles or 30.6% in 2020. Research focuses include wind resource assessment, blade design, monitoring technology, structural dynamic response analysis, integrated design, etc., demonstrating China's efforts in exploration and innovation in these aspects. China's influence and international academic standing in the OWP industry have grown.

China has made advanced achievements in the wind power industry, such as control technology, large-scale wind turbine design, coordinated control technology of wind farm clusters, and friendly grid connection of wind power. Other technologies such as offshore engineering equipment and high voltage flexible DC transmission are also in tremendous progress. From 2016 to 2020, the National Natural Science Foundation of China has funded 73 projects related to OWP, which is over four times higher than that during between 2012 and 2016. Table 2 summarize the key technologies developed in China from 2010 to 2020. These research projects and the national R&D plan mainly focus on the R&D and testing technology of large-capacity offshore wind turbines, support structure optimization, floating structure design, offshore wind farm group planning, friendly grid connection technology, OWP intelligent operation, and maintenance technology, etc. As the investments of the state, developers, enterprises, and universities put into the OWP industry are increasing yearly, the related theoretical and achievements and technologies will also show a good growth trend.

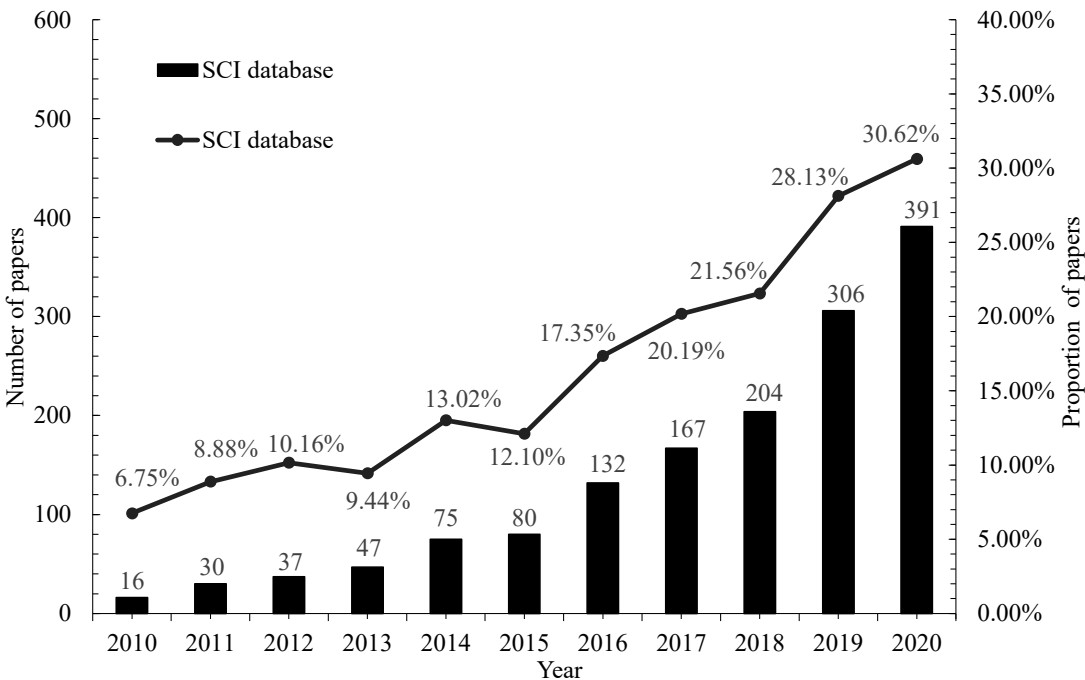

**Figure 9.** Chinese scholar articles collected in the SCI database.

*5.2. Standardization of OWP in China*

Since the late 1980s, the IEC/TC88 Technical Committee has begun to organize the preparation of international standards for wind turbines. IEC published the first offshore wind turbine standard in 2009. IEC regularly updates original standards and adds new standards based on the application of standards and the growth of the wind power industry. Currently, major OWP markets such as Denmark, Germany, and the UK use the IEC 61400 series wind turbine standards, which include wind turbine design requirements, blade testing, power characteristic testing, load measurement, etc. They have combined IEC 61400 with their national standards to further optimize and supplement their national standards.

The National Standardization Management Committee in China published the construction specifications for OWP projects in 2010 and outlined technical requirements for construction, transportation, infrastructure requirements, power generation equipment installation and project management, etc. The first national standard for offshore wind farms (GB/T 51308-2019), published by China Energy Construction Group Planning and Design Co., Ltd., has been in effect since 1 October 2019, following nearly ten years of development. Regarding the OWP certification system, the construction of the testing and certification system for OWP equipment has always been highly valued as a crucial component of quality assurance in Denmark and Germany, and relevant standards such as IEC WT01 are set for certification. However, China still needs to establish a complete set of "technology R&D, testing certification, manufacturing, operation feedback" systems [59].

China should promote the creation of national standards that reflect local regional conditions and enhance the testing and certification process. Meanwhile, China needs to actively engage in the creation of international OWP standards to integrate its standards with those of other countries and lead the development of OWP globally.

*5.3. Parity Process of China's Offshore Wind Industry*

China's OWP pricing has experienced the approved-price stage from 2008 to 2014. Since then, a number of government bodies have announced numerous new pricing rules in conjunction with the revival of the OWP business. The newly approved OWP guidance price was changed to 0.8 yuan per kWh in 2019 and to 0.75 yuan per kWh in 2020. This price is part of the annual management of financial subsidies. China's OWP has entered the

competitive age of bidding with the determination of the feed-in tariff for newly approved OWP projects, which shall not be higher than the above guidance price.

**Table 2.** Key technologies for wind power developed in China from 2010 to 2020.

| | Technology | Technical Points | Research Projects [58] | National R&D | Importance |
|---|---|---|---|---|---|
| 1 | Refined assessment of offshore wind resources | • Wind resources analysis<br>• Wind turbine layout optimization | 3 | | Important |
| 2 | Floating foundation design | • Mooring systems<br>• Dynamic cables<br>• Integrated analysis of floating structures | 12 | | Very important |
| 3 | Intelligent operation and maintenance | • Condition monitoring and fault diagnosis<br>• Intelligent decision-making method<br>• Operation optimization of wind farm | 8 | | Very important |
| 4 | Structural design | • Structural dynamic response<br>• Foundation design<br>• Integrated design of wind turbine and foundation | 49 | | Extremely important |
| 5 | Generator design | • Permanent magnet synchronous generator<br>• Large bearings<br>• High power density converter | 8 | 3 | Very important |
| 6 | Blade design | • Lightweight blade material<br>• Management optimization of blade materials<br>• Coupling of blade and pitch system | 3 | 1 | Important |
| 7 | Power system | • Flexible DC transmission<br>• System stability analysis<br>• System control technology | 22 | 2 | Extremely important |
| 8 | Engineering geological survey | • Advanced geotechnical test | 0 | | Important |
| 9 | Construction technology | • Construction safety<br>• Construction equipment manufacturing | 1 | 1 | Important |

In January 2020, the government issued "several opinions on promoting the healthy development of non-hydropower renewable energy power generation", which clearly pointed out that new OWP projects will not be eligible for national financial subsidies from 2020; instead, local governments will provide support. Offshore wind farm projects that have been approved (put on record) according to regulations or have all wind turbines connected to the grid before 31 December 2021 will be incorporated into the national financial subsidies in accordance with the corresponding price policy. However, per the status quo and existing industrial development policies, China's OWP will obtain no subsidies after 2025. This legislative change will speed up China's transition to parity in the offshore wind sector and provide significant obstacles for the overall sector. To ensure the healthy development of China's OWP, it is crucial to figure out how to obtain local government subsidies and quicken technology advancements to attain parity.

*5.4. Future Development of China's Offshore Wind Industry*

Thanks to independent innovation and references from the experience of global OWP development, China has steadily realized the autonomy of technology, and its OWP development intensity has surpassed that of the rest of the world. Even though China is making technological strides, some crucial technologies, such as large-capacity wind turbine manufacturing, floating structure design, and OWP hydrogen production technology, are still behind some other countries. The following technologies are important to achieve significant cost reduction: (1) accurate analysis and optimal utilization of wind resource exploitation; (2) customized wind farm development and wind turbine; (3) integrated design of (floating) large-capacity wind turbines; (4) integration of OWP system; (5) intelligent construction and O&M in the full life cycle.

Full life cycle asset management is one of the practical solutions for intelligent construction and O&M. The characteristics of offshore wind farm assets include poor accessibility, strong correlation, high operation and maintenance costs, and long operating life. Based on the management concept of asset-centered full life cycle, closed-loop, feedback, and intelligent management, the OWP enterprises should develop an asset management tool and confirm the project asset management goals and index systems, along with WBS (Work Breakdown Structure) and CBS (Cost Breakdown Structure), in order to build an offshore wind project ABS (Asset Breakdown Structure). The ABS decomposes the work tasks of assets of each full life cycle stage into a minimum work unit that can be evaluated, with schedule plan, resource requirement, cost budget, and risk management plan, ensuring the practical implementation of asset management in the entire life cycle of OWP projects. By using the ABS-WBS-CBS coupling structure, managers could integrate the work task effects of each stage to evaluate the performance of assets via earned value analysis, net present value bias analysis, and risk analysis so as to optimize the asset management decision and enhance the capacity for asset value creation.

## 6. Conclusions

The key technologies for offshore wind farms, including site selection, design, construction, and O&M in China, are reviewed in this study. The challenges are examined based on the projects that are currently under construction and those that are planned. Suggestions are made for future development in China. The state-of-the art methods for wind energy resource assessment, site surveying, and environmental protection technologies are summarized. Currently, China is conducting research and building demonstration projects to optimize wind turbine layouts considering typhoons and to explore marine resources comprehensively. With the limited site survey, extra consideration must be given to the selection or development of marine geological survey methods for superior geological information for trustworthy design and safe construction.

Aiming to construct large-scale wind turbines, China has developed the capability of manufacturing 5 MW offshore wind turbines and will continue to study the manufacturing technology for essential components of 10 MW+ PMSG wind turbines, such as high-performance bearings, IGBT, and high-end sensors. Additionally, as the exploration of OWP approaches into deeper seabed, China is preparing the commercialization of floating structures and flexible DC transmission technology for the next boom in OWP.

After the fast growth before 2022, China has won great construction achievements. However, the shortage of equipment resources limits the efficiency improvement of construction and O&M. More professional vessels and equipment are expected to be used for offshore wind farm projects in the near future, which will prolong the construction window period and improve the safety and efficiency of construction. Advanced equipment will also promote O&M from the traditional into the intelligent mode.

Although China's OWP industry has made great strides in lowering costs and improving the power generation efficiency of OWP projects, challenging marine conditions (typhoons, earthquakes, ice, deep soft soils, etc.) and COVID-19 cause additional difficulties

to this sector. Government support and direction are essential during such a unique time, from planning to technological advancements and demonstrations.

**Author Contributions:** Conceptualization, Q.F. and P.L.; formal analysis, X.W. investigation, X.W.; resources, Q.F. and P.L.; writing, review and editing, X.W., J.Y., Q.F. and P.L.; supervision, Q.F., P.L. and J.Y.; project administration, X.L. and H.H. All authors have read and agreed to the published version of the manuscript.

**Funding:** This research was funded by China Huaneng Group Co., Ltd., under grant number HNKJ19-H16, HNKJ20-H53, and the Fund Program of State Key Laboratory of Hydroscience and Engineering, under grant number 2022-KY-05.

**Institutional Review Board Statement:** Not applicable.

**Informed Consent Statement:** Not applicable.

**Data Availability Statement:** Only public data duly referenced are used in the present study.

**Conflicts of Interest:** The authors declare no conflict of interest.

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
