# Peer review of "A Review of the Development of Key Technologies for Offshore Wind Power in China"

_jmse, doi:10.3390/jmse10070929_

Round 1

Reviewer 1 Report

Dear authors,

The topic of your review paper “A Review of Development of Core Technology for Offshore Wind Power in China” is interesting and relevant. I have the following comments and questions:

I believe the topics and scope of the manuscript are interesting and relevant for the wind industry. However, I found that many of the topics were discussed in a superficial and sometimes outdated mode. Most of the references are older than 5 years and therefore don’t capture current development status of offshore wind energy. I would suggests increasing the number of more recent papers in your review paper and disusing in more detail the most important challenges and opportunities for offshore wind energy.

One of the most important current challenges for the future growth of wind energy is the disruption of supply chains caused by the Covid-19 pandemic. The supply chain challenges are mentioned superficially in your manuscript. I would suggest developing a more robust supply chain section indicating relevant challenges, potential solutions and opportunities for offshore wind growth.

Lines 90 – 92 indicate that “The main objective of this paper is to point out the technical innovation direction to reduce the cost and increase the efficiency of offshore wind power through discussing the core technologies of offshore wind power industry, as shown in Fig.3.” I believe this objective is very general and lacks specificity. Furthermore, given the scope of the review paper I believe that there is not a clear connection between the topics reviewed and the proposed objective.  It is not clear why Figure 3 helps to explain the objective of the research. This needs to be further explained and expanded.

I would suggest for the novelty and scientific contribution of the research to be strengthened and adjusted to connect with the topics discussed on your review. The discussion and strengthening of the scientific contribution will allow readers to better understand the relevance of your review and the usefulness that can have for wind industry.

Many of the figures (1, 2, 3. 4, 5, 8, 9, 12) are blurry and difficult to read. The resolution and quality of the images needs to be significantly improved. Additionally, the Copyright of the Figures that contain photographs or images needs to be verified. Were the photos taken by the authors, the images created by the authors or do you have Copyright authorization to publish them in this paper? This is very important to verify.

Abstract and conclusion need to be rewritten. In their current form they are too general and do not allow the reader to have a good overview of the review paper.

The English grammar and construction need to be reviewed and improved. I would suggest for the final version of the manuscript to be reviewed by a proficient English native speaking editor.

Author Response

Response to the Reviewers’ Comments

Manuscript ID: jmse-1788421

Title: A Review of Development of Core Technology for Offshore Wind Power in China

Author(s):Qixiang Fana,*, Xin Wangb, Jing Yuanb, Xin Liuc, Hao Hud and Peng Linb,*

Dear Reviewer:

Thank you for investing your valuable time in reviewing the manuscript and providing comments and suggestions. Those comments were found convenient and helpful for improving the content and quality of the manuscript. According to your comments, we have revised the manuscript. The changes in the manuscript have been marked up using the “Track Changes” in order to facilitate their identification.

Comment1: I believe the topics and scope of the manuscript are interesting and relevant for the wind industry. However, I found that many of the topics were discussed in a superficial and sometimes outdated mode. Most of the references are older than 5 years and therefore don’t capture current development status of offshore wind energy. I would suggests increasing the number of more recent papers in your review paper and disusing in more detail the most important challenges and opportunities for offshore wind energy.

Reply: Thanks for your constructive suggestion! We have renewed a few more recent papers and accordingly discussions as follows:

1) Section 2.1, Page 5, Lines 164-174.

2) Section 3.1.4, Page 12, Lines 412-414.

3) Section 3.3.2, Page 15, Lines 543-553.

4) Section 4.2.1, Page 19, Lines 698-710.

5) Section 4.2.1, Page 20, Lines 720-730..

6) Section 4.2.2, Page 20, Lines751-760.

The references are listed below:

(Abdelbaky, Liu, Kong, & Ieee, 2019; Du et al., 2017; Pillai, Chick, Khorasanchi, Barbouchi, & Johanning, 2017; Sierra-Garcia & Santos, 2021; Silva & Estanqueiro, 2013; Strbac, Greiwe, Hoffmann, Cormier, & Lutz, 2022; Sun, Yang, & Gao, 2019; Zou, Wei, Feng, & Zhou, 2022)

Abdelbaky, M. A., Liu, X., Kong, X., & Ieee. (2019). Wind Turbines Pitch Controller using Constrained Fuzzy-Receding Horizon Control. Paper presented at the 31st Chinese Control And Decision Conference (CCDC), Nanchang, PEOPLES R CHINA.

Du, M., Yi, J., Guo, J., Cheng, L., Ma, S., & He, Q. (2017). Review on reliability centred maintenance  strategy and applications to offshore wind farm operation and maintenance. Power System Technology, 41(7), 2247-2254.

Pillai, A. C., Chick, J., Khorasanchi, M., Barbouchi, S., & Johanning, L. (2017). Application of an offshore wind farm layout optimization methodology at Middelgrunden wind farm. Ocean Engineering, 139, 287-297. doi:10.1016/j.oceaneng.2017.04.049

Sierra-Garcia, J. E., & Santos, M. (2021). Improving Wind Turbine Pitch Control by Effective Wind Neuro-Estimators. Ieee Access, 9, 10413-10425. doi:10.1109/access.2021.3051063

Silva, N., & Estanqueiro, A. (2013). Impact of Weather Conditions on the Windows of Opportunity for Operation of Offshore Wind Farms in Portugal. Wind Engineering, 37(3), 257-268. doi:10.1260/0309-524x.37.3.257

Strbac, A., Greiwe, D. H., Hoffmann, F., Cormier, M., & Lutz, T. (2022). Piloted Simulation of the Rotorcraft Wind Turbine Wake Interaction during Hover and Transit Flights. Energies, 15(5). doi:10.3390/en15051790

Sun, H., Yang, H., & Gao, X. (2019). Investigation into spacing restriction and layout optimization of wind farm with multiple types of wind turbines. Energy, 168, 637-650. doi:10.1016/j.energy.2018.11.073

Zou, C., Wei, R., Feng, J., & Zhou, Y. (2022). Development status and application prospect of VSC-HVDC. Southern Power System Technology, 16(3), 1-7.

Comment2: One of the most important current challenges for the future growth of wind energy is the disruption of supply chains caused by the Covid-19 pandemic. The supply chain challenges are mentioned superficially in your manuscript. I would suggest developing a more robust supply chain section indicating relevant challenges, potential solutions and opportunities for offshore wind growth.

Reply: Thank you for the suggestion. We have added a part of the discussion on China’s supply chains during the Covid-19 pandemic in Section 3.4.

Please see Section 3.4, Page 15, Lines 555-589.

Comment3: Lines 90 – 92 indicate that “The main objective of this paper is to point out the technical innovation direction to reduce the cost and increase the efficiency of offshore wind power through discussing the core technologies of offshore wind power industry, as shown in Fig.3.” I believe this objective is very general and lacks specificity. Furthermore, given the scope of the review paper I believe that there is not a clear connection between the topics reviewed and the proposed objective. It is not clear why Figure 3 helps to explain the objective of the research. This needs to be further explained and expanded.

Reply: Thanks for the constructive suggestion! We have revised the ending of the introduction (including Figure 2 and Figure 3) according to your valuable suggestions.

Please see Section 1, Page 3, Lines 93-114.

Comment4: I would suggest for the novelty and scientific contribution of the research to be strengthened and adjusted to connect with the topics discussed on your review. The discussion and strengthening of the scientific contribution will allow readers to better understand the relevance of your review and the usefulness that can have for wind industry.

Reply: Thanks for your approval of this paper. We have revised the whole article and now the quality of the manuscript has been improved according to your valuable comments.

Comment5: Many of the figures (1, 2, 3. 4, 5, 8, 9, 12) are blurry and difficult to read. The resolution and quality of the images needs to be significantly improved. Additionally, the Copyright of the Figures that contain photographs or images needs to be verified. Were the photos taken by the authors, the images created by the authors or do you have Copyright authorization to publish them in this paper? This is very important to verify.

Reply: Thanks for your valuable suggestion. All figures have been replaced with higher quality Encapsulated PostScript (EPS). For the Copyright, we would like to state that figures 1, 2, 3, 4, 5, 6, 9, 10, 11 are created by authors, and figure 7 and figure 8 are photos from authors’ cooperating enterprises that we can provide Copyright authorization if needed.

Please see Section1, Pages 3-5; Section 2.1, Page 6; Section 2.4, Pages 8-Page 9; Section 4.1, Pages 17-18; Section 4.2, Page 21; Section 4.3, Page 21; Section 5.1, Page 23.

Comment6: Abstract and conclusion need to be rewritten. In their current form they are too general and do not allow the reader to have a good overview of the review paper.

Reply: Thanks for the constructive suggestion! We have revised the abstract and conclusion according to your valuable suggestion.

Please see Abstract, Page 1; Section 6, Page 25.

Comment7: The English grammar and construction need to be reviewed and improved. I would suggest for the final version of the manuscript to be reviewed by a proficient English native speaking editor.

Reply: Thanks for your suggestion. We will have our manuscript checked by a native English-speaking colleague.

Reviewer 2 Report

The paper concerns the performed review of offshore wind farm's core technologies, including wind turbine design problems, foundation structure design, construction problems, and maintenance for offshore wind power. The main objective of this paper is to highlight the technical innovation direction to reduce the cost and increase the efficiency of offshore wind power by discussing the core technologies of the offshore wind power industry. The problem is interesting; however, I have some comments.

- It seems to me that the title should be modified. It should indicate an overview of the development of core technology for offshore wind power. In the current form, the article is limited to the local scale (single country). The journal has an international scale, so in my opinion the review should be done in a global area, using China as an example, among others. As it is presently formulated, I am afraid that this will only limit the number of potential readers.

- I propose to modify the abstract. After reading it, it is difficult to deduce what problems were solved by the authors and what goals were achieved.

- The quality of the figures should be improved.

- The authors should clearly emphasize the purpose of the review conducted and indicate what the paper contributes to science.

Author Response

Response to the Reviewers’ Comments

Manuscript ID: jmse-1788421

Title: A Review of Development of Core Technology for Offshore Wind Power in China

Author(s):Qixiang Fana,*, Xin Wangb, Jing Yuanb, Xin Liuc, Hao Hud and Peng Linb,*

Dear Reviewer:

Thank you for investing your valuable time in reviewing the manuscript and providing comments and suggestions. Those comments were found convenient and helpful for improving the content and quality of the manuscript. According to your comments, we have revised the manuscript. The changes in the manuscript have been marked up using the “Track Changes” in order to facilitate their identification.

Comment1: It seems to me that the title should be modified. It should indicate an overview of the development of core technology for offshore wind power. In the current form, the article is limited to the local scale (single country). The journal has an international scale, so in my opinion the review should be done in a global area, using China as an example, among others. As it is presently formulated, I am afraid that this will only limit the number of potential readers.

Reply: Thanks for your kind reminder. We agree that offshore wind power is a global issue, but we feel a paper that specifically discussing the China’s situation is still a valuable piece of work, which may attract many readers.  

Firstly, this paper reviews the research progress of core technologies in the global offshore wind power industry, discusses the challenges currently faced by China’s offshore wind power industry, and proposes its future development direction. Due to the different natural conditions, technical levels, and policy support in various countries, the main dilemma can be very different in different countries, so making recommendations from the respective of global industries without regard to regional characteristics may lack credibility. For example, the development of the offshore wind power industry in China is nearly 15 years later than that in Europe, and it still needs to localize key components and significantly reduce the project costs. Meanwhile, China’s offshore wind power development is considerably driven by subsidy policies and is currently concentrated in offshore waters due to the harsh marine environment, such as threatens from typhoons, earthquakes, sea ice, and soft soil, while the European offshore wind power industry has mainly focused on deep-sea and comprehensive marine development, so the main concerns of offshore wind power development in China and Europe are obviously different.

Secondly, China has now become the largest country in the world in terms of installed capacity of offshore wind power, which has attracted the attention of offshore wind power practitioners all over the world. The development history and future development direction of China's offshore wind power industry can also serve as a reference for some countries that are about to open the era of offshore wind power.

Comment2: I propose to modify the abstract. After reading it, it is difficult to deduce what problems were solved by the authors and what goals were achieved.

Reply: Thanks for the constructive suggestion! We have revised the abstract according to your valuable suggestion.

Please see Abstract, Page 1.

Comment3: The quality of the figures should be improved.

Reply: Thanks for your valuable suggestion. All figures have been replaced with higher quality Encapsulated Post Script (EPS).

Please see Section1, Pages 3-5; Section 2.1, Page 6; Section 2.4, Pages 8-Page 9; Section 4.1, Pages 17-18; Section 4.2, Page 21; Section 4.3, Page 21; Section 5.1, Page 23.

Comment4: The authors should clearly emphasize the purpose of the review conducted and indicate what the paper contributes to science

Reply: Thanks for your approval of this paper. We have revised the ending of the introduction (including Figure 2 and Figure 3) and the conclusion of the manuscript according to your valuable comment.

Please see Section 1, Page 3, Lines 93-114; Section 6, Page 26.

Round 2

Reviewer 1 Report

Many of the review comments were addressed by the authors.

I still have some comments,

Some images are still low quality. The figures need to be improved.

It should be indicated which pictures were taken by the authors and indicate the proper copyright for the pictures taken by other persons. 

I still suggest improvement in the edition and English grammar and construction.  

Author Response

Response to the Reviewers’ Comments

Manuscript ID: jmse-1788421

Title: A Review of Development of Core Technology for Offshore Wind Power in China

Author(s):Qixiang Fana,*, Xin Wangb, Jing Yuanb, Xin Liuc, Hao Hud and Peng Linb,*

Dear Reviewer:

Thank you again for investing your valuable time in reviewing the manuscript and providing comments and suggestions. Those comments were found convenient and helpful for improving the content and quality of the manuscript. According to your comments, we have revised the manuscript. The changes in the manuscript have been marked up using the “Track Changes” in order to facilitate their identification.

Comment1: Some images are still low quality. The figures need to be improved. It should be indicated which pictures were taken by the authors and indicate the proper copyright for the pictures taken by other persons.

Reply: Thanks for your kind reminder. We have deleted Figure 7 and figure 8 in the previous version of manuscript for lacking original photographs. So we would like to state that all drawings are created by authors and we have uploaded high-quality figures in the attachment. Besides, It is said by Editor Zhang that the quality of figures was decreased automatically while converting to pdf file.

Comment2: I still suggest improvement in the edition and English grammar and construction.

Reply: Thanks for your valuable suggestion. We have carefully checked and improved the English writing in the revised manuscript.

Reviewer 2 Report

Thank you for sending the answers to my questions. All comments and concerns have been sufficiently addressed. The manuscript has now been sufficiently improved. I suggest only, if possible, to improve the quality of some drawings and possibly to improve the English language.

Author Response

Response to the Reviewers’ Comments

Manuscript ID: jmse-1788421

Title: A Review of Development of Core Technology for Offshore Wind Power in China

Author(s):Qixiang Fana,*, Xin Wangb, Jing Yuanb, Xin Liuc, Hao Hud and Peng Linb,*

Dear Reviewer:

Thank you again for investing your valuable time in reviewing the manuscript and providing comments and suggestions. Those comments were found convenient and helpful for improving the content and quality of the manuscript. According to your comments, we have revised the manuscript. The changes in the manuscript have been marked up using the “Track Changes” in order to facilitate their identification.

Comment1: if possible, to improve the quality of some drawings.

Reply: Thanks for your kind reminder. We have deleted Figure 7 and figure 8 in the previous version of manuscript for lacking original photographs. So we would like to state that all drawings are created by authors and we have uploaded high-quality figures in the attachment. Besides, It is said by Editor Zhang that the quality of figures was decreased automatically while converting to pdf file.

Comment2: possibly to improve the English language.

Reply: Thanks for your valuable suggestion. We have carefully checked and improved the English writing in the revised manuscript.
